# Lunaemycins, New Cyclic Hexapeptide Antibiotics from the Cave Moonmilk-Dweller *Streptomyces lunaelactis* MM109^T^

**DOI:** 10.3390/ijms24021114

**Published:** 2023-01-06

**Authors:** Loïc Martinet, Aymeric Naômé, Lucas C. D. Rezende, Déborah Tellatin, Bernard Pignon, Jean-Denis Docquier, Filomena Sannio, Dominique Baiwir, Gabriel Mazzucchelli, Michel Frédérich, Sébastien Rigali

**Affiliations:** 1InBioS—Centre for Protein Engineering, Institut de Chimie B6a, University of Liège, B-4000 Liege, Belgium; 2Hedera-22, Boulevard du Rectorat 27b, B-4000 Liege, Belgium; 3Dipartimento di Biotecnologie Mediche, University of Siena, Viale Bracci 16, 53100 Siena, Italy; 4GIGA Proteomics Facility, University of Liege, B-4000 Liege, Belgium; 5Mass Spectrometry Laboratory, MolSys Research Unit, University of Liege, B-4000 Liege, Belgium; 6Laboratory of Pharmacognosy, Center of Interdisciplinary Research on Medicines (CIRM), University of Liege, B-4000 Liege, Belgium

**Keywords:** antibiotics, cryptic metabolites, natural compounds, Anti-MRSA, moonmilk, cave microbiology, genome mining, piperazic acid, NRPS, molecular networking

## Abstract

*Streptomyces lunaelactis* strains have been isolated from moonmilk deposits, which are calcium carbonate speleothems used for centuries in traditional medicine for their antimicrobial properties. Genome mining revealed that these strains are a remarkable example of a *Streptomyces* species with huge heterogeneity regarding their content in biosynthetic gene clusters (BGCs) for specialized metabolite production. BGC 28a is one of the cryptic BGCs that is only carried by a subgroup of *S*. *lunaelactis* strains for which in silico analysis predicted the production of nonribosomal peptide antibiotics containing the non-proteogenic amino acid piperazic acid (Piz). Comparative metabolomics of culture extracts of *S*. *lunaelactis* strains either holding or not holding BGC 28a combined with MS/MS-guided peptidogenomics and ^1^H/^13^C NMR allowed us to identify the cyclic hexapeptide with the amino acid sequence (D-Phe)-(L-HO-Ile)-(D-Piz)-(L-Piz)-(D-Piz)-(L-Piz), called lunaemycin A, as the main compound synthesized by BGC 28a. Molecular networking further identified 18 additional lunaemycins, with 14 of them having their structure elucidated by HRMS/MS. Antimicrobial assays demonstrated a significant bactericidal activity of lunaemycins against Gram-positive bacteria, including multi-drug resistant clinical isolates. Our work demonstrates how an accurate in silico analysis of a cryptic BGC can highly facilitate the identification, the structural elucidation, and the bioactivity of its associated specialized metabolites.

## 1. Introduction

*Streptomyces* and other actinomycetes are Gram-positive filamentous bacteria that offered most of the molecules of microbial origin that are now used in human and animal therapy [1,2]. The urge for new structural leads for both the agro-food and pharmaceutical fields has revitalized the interest for bioprospection of microorganisms dwelling in the most diverse and extreme environmental niches [3]. This quest for metabolite-producing bacteria in environments far from the ecological niches where they were mainly/originally isolated (such as rich organic soils for actinomycetes) is motivated by the idea that they would possess genetic features adapted to these specific and unusual environments, allowing them to produce uncommon ‘specialized’ metabolites [4].

Caves, being inorganic and extreme oligotrophic environments, were initially considered as inhospitable places for streptomycetes and other microorganisms programmed for primarily consuming nutrients derived from the plant decomposing organic matter. Instead, this phylum is surprisingly omnipresent, displaying relative abundance and diversity depending on the type of cave and speleothem [5,6,7,8,9,10]. One such geological formation that was previously unexpected to house a huge diversity of actinomycetes is the so-called moonmilk, a soft white carbonate deposit in limestone caves mainly consisting of microfibers of calcite [11,12,13]. Even within the same cave, the diversity of actinomycetes can widely differ from one moonmilk deposit to another [6]. More astonishingly, these bacteria do not only inhabit moonmilk to benefit from the nutrients carried by the water percolating from the surface, but they also actively participate in the creation of the mineral structure itself [12,14,15,16,17,18].

The decision of our laboratory to search for antibiotic-producing bacteria in moonmilk deposits originated from evidence of hundreds of years of mining activities for the use of this speleothem as powder with antimicrobial properties [13,19]. The first scanning electron micrographs of the calcite fibers of moonmilk also revealed filamentous bacteria further supporting the presence of microorganisms notorious as ‘antibiotic-makers’ [14,15,16,18]. Indeed, each actinomycete we isolated from three different moonmilk deposits of the cave ‘la grotte des collemboles’ (Comblain-au-Pont, Belgium) displayed antibacterial and/or antifungal activities [9,10]. A *Streptomyces* species that we found in all moonmilk deposits of the cave is *Streptomyces lunaelactis* with isolate MM109^T^ as the archetype strain [9,10,20,21]. *S*. *lunaelactis* MM109^T^ is the first fully characterized and completely sequenced *Streptomyces* strain originating from these speleothems [20,21,22]. Genome mining of the 18 *S*. *lunaelactis* strains revealed 42 biosynthetic gene clusters (BGC) with only 18 of them being conserved among all strains, thereby providing a remarkable example of a huge specialized metabolism heterogeneity within a single species [22].

One of our aims in investigating cave microbiology is to identify the genetic and physiological features that make soil-dwelling bacteria to adapt to inorganic mineral environments [15]. Some of these adjustments must be part of their specialized metabolism used to capture essential nutrients and compete with other microorganisms or instead contribute to the survival of the entire microbial community. In our effort to identify the metabolites produced by moonmilk-dwelling actinomycetes, interesting findings came to light, such as the discovery of the BGC responsible for the hallmark green pigmentation of *S*. *lunaelactis* strains, which revealed a unique example of a BGC involved in the production of two structurally and functionally different metabolites, namely the bagremycin antibiotics and the iron chelating molecules that generate ferroverdins [22,23,24].

Herein, we report how genome mining guided MS/MS-based networking and peptidogenomics allowed the discovery, the elucidation of the structure, and the biosynthetic pathway of lunaemycins, new, potent antibiotics produced by *S*. *lunaelactis* MM109^T^. Lunaemycins are new, piperazic acid-containing cyclic hexapeptides able to inhibit the growth of all tested Gram-positive bacteria, including various multi-drug resistant isolates, thereby in part explaining the success of this species at colonizing extremely competitive environments.

## 2. Results and Discussion

### 2.1. Sequence Analysis of BGC 28a and Predicted Structure of Its Associated Natural Product

Mining the genome of *S*. *lunaelactis* MM109^T^ [21] with antiSMASH identified BGC 28a, a ~69 kb non-ribosomal peptide synthetase (NRPS)-type BGC consisting of 39 open reading frames (ORFs) from nucleotide positions 3124 to 72,040 on the linear plasmid pSLUN1a (Figure 1 and Table 1). The *hmt* gene cluster of *Streptomyces himastatinicus* producing the antitumor antibiotic himastatins [25,26] is the most similar BGC of the MIBiG database [27] in terms of number of homologous genes (Appendix A and Table 1). Further sequence analysis highlighted three other known BGCs sharing common features with BGC 28a, namely the *ktz* gene cluster producing the antifungal kutznerides in *Kutzneria* sp. 744 [28], the *sten* gene cluster associated with the production of the antimicrobial stenothricins in *Streptomyces roseosporus* NRRL 15998 [29], and the *art* BGC for the antitumor antibiotic aurantimycins produced by *Streptomyces aurantiacus* JA 4570 [30]. The synteny and conserved gene content of these four BGCs is relatively low with *hmt*, *ktz*, *art*, and *sten* gene clusters sharing only nine, seven, six, and four homologous genes with the 39 ORFs of BGC 28a, respectively (Appendix A). Himastatins, kutznerides, stenothricins, and aurantimycins are all cyclic depsipeptides synthesized by NRPS or hybrid NRPS-PKS machineries [31,32,33,34], suggesting that the main product of BGC 28a should partially share some structural features and/or amino acid building blocks with these natural products.

Closer sequence analysis of the core and additional biosynthetic genes of BGC 28a allowed us to propose a biosynthetic pathway and predict the amino-acid building blocks and backbone of its main product (Figure 1). First, as observed in the BGCs of kutznerides and himastatins, BGC 28a contains the two genes required for the biosynthesis of the non-proteinogenic amino acid piperazic acid (Piz) from L-ornithine (Orn). The first reaction is performed by the product of *lun16* encoding the flavin- and oxygen-dependent L-ornithine N-hydroxylase (SLUN_ RS38485 homologous to HmtM and KtzI, Figure 1, Table 1) to give L-N^5^-OH-ornithine [35], which will be further converted to L-Piz by the product of *lun17* (SLUN_ RS38490 homologous to HmtC and KtzT, Figure 1, Table 1) encoding the heme-dependent piperazate synthase [36]. Interestingly, genes encoding piperazate synthase are present in most BGCs for Piz-containing molecules [37,38], suggesting that at least one NRPS module of BGC 28a is expected to recruit Piz as building block. Next, as observed in the stenothricin gene cluster [29], BGC 28a contains a transcription unit with four ORFs whose products encode the enzymes involved in the five steps of Orn biosynthesis from glutamic acid and glutamine precursors (Figure 1, Table 1). These genes/proteins are *lun14* for the bifunctional N^2^-acetyl-L-ornithine:L-glutamate N-acetyltransferase (SLUN_ RS38475 homologous to StenL, Figure 1, Table 1), *lun13* for the N-acetylglutamate kinase (SLUN_RS38470 homologous to StenK and HmtB, Figure 1, Table 1), *lun11* for the N-acetyl-gamma-glutamyl-phosphate reductase (SLUN_RS38460 homologous to StenM, Figure 1, Table 1), and *lun12* for the N^2^-acetyl-L-ornithine:2-oxoglutarate aminotransferase (SLUN_RS38465 homologous to StenJ, Figure 1, Table 1). BGC 28a thus contains all genes required for the enzymatic conversion of glutamic acid and glutamine to L-Piz, further supporting the recruitment of this amino acid by the NRPS machinery. 

Regarding the core biosynthetic genes, three ORFs encode NRPSs with complete biosynthetic modules. The sequential order of these modules in peptide synthesis (initiation, elongation, and termination modules) is predicted based on the composition of their catalytic domains (Figure 1). The biosynthesis would start with Lun23 (SLUN_ RS38520, providing the initiation module 1 and the ‘inactive’ module 2), followed by Lun20 (SLUN_ RS38505, elongation module 3), and finally Lun22 (SLUN_RS38515, elongation and termination modules 4, 5, 6, and 7) (Figure 1). All together these NRPSs form a megasynthetase organized into seven modules, for which the detailed analysis of the specificity-conferring sequence of each adenylation (A) domains with SANDPUMA [39] allowed the prediction of the building blocks recruited by the NRPS machinery (Figure 1).

Module 1 in Lun23 contains three domains (A-T-E) and is predicted to incorporate L-Phe (DAWTVAAVCK as 10-aa code), which would be converted to D-Phe by the epimerization (E) domain. Module 2 in Lun23 also contains three domains (C-A-T), but prediction software tools did not propose specificity to any amino acid and suggest that this module would be inactive (DVASLAAYAK as 10-aa code). Module 3 in Lun20 contains three domains (C-A-T) and is predicted to recruit L-Ile (DAFFLGVTYK as 10-aa code). The four following modules (4 to 7), all included in Lun22, are predicted to incorporate four Piz residues, two D-Piz via modules 4 (C-A-T-E) and 6 (C-A-T-E) that both contain an epimerization domain, and two L-Piz via module 5 (C-A-T) and module 7 (C-A-T-TE). Indeed, the 10 aa code sequence conferring amino acid substrate specificity for Lun22 (DVFSVAAYAK as 10-aa code for modules 4, 5, 6, and 7) is identical to the one of HmtL-A1 (DVFSVAAYAK), and highly similar to those of KtzH-A1 (DVFSVGPYAK as 10-aa code, 8/10), ArtF-A1 (DVFSVASYAK as 10-aa code, 9/10), and ArtG-A2 (DVFTVAAYAK as 10-aa code, 9/10) which were reported to recognize and activate Piz in previous studies [25,28,30,37,38]. Finally, module 7 would cyclize and release the final hexapeptide via its thioesterase (TE) domain as shown for the biosynthesis of himastatins and kutznerides where the last TE domain of HmtL and KtzH, the homologues of Lun22, perform the macrocyclization of their respective final peptide [25,28].

In conclusion, the NRPS machinery of BGC 28a is predicted to generate an hexapeptide with the sequence D- Phe, L-Ile, D-Piz, L-Piz, D-Piz, L-Piz as backbone of its main product that, with or without cyclization, would have monoisotopic masses of 708.4071 Da (C_35_H_52_N_10_O_6_) and 726.4177 Da (C_35_H_54_N_10_O_7_), respectively.

### 2.2. Structure Elucidation of Lunaemycins

#### 2.2.1. MS/MS-Based Networking and Peptidogenomics Guided Genome Mining

A peptidogenomic approach where genome mining guides the tandem mass spectrometry (MS/MS)-based molecular identification of high-resolution mass spectrometry (HRMS) data was used to identify compounds produced by BGC 28a. The predicted building-blocks can be used to screen the MS/MS fragments in order to find neutral or charged losses of amino-acids. Although BGC 28a is predicted to produce a cyclized hexapeptide of a monoisotopic mass of 708.4071 Da, we cannot exclude at this stage either modifications of one or more building blocks before their utilization by the NRPS machinery and/or post modifications of the generated hexapeptide. Therefore, the MS/MS spectra of molecular ions ranging from m/z 700 to 800 Da have been manually analyzed to identify tag fragments corresponding to the three different amino acids predicted to be incorporated by the six NRPS modules. Finally, prior to performing the comparative metabolomic analysis, genome mining of the eighteen isolated *S*. *lunaelactis* strains revealed a genetic characteristic that is crucial for the identification of the products of BGC 28a. Indeed, BGC 28a is present in only three *S*. *lunaelactis* strains, i.e., those that possess pSLUN1a, whereas all the other strains with the linear plasmid pSLUN1b instead comprise BGC 28b [22] (Figure 2a). *S*. *lunaelactis* strains that possess pSLUN1b are therefore natural variants/mutants non-producing the searched compound(s) of BGC 28a, and their culture extract can be considered as negative control.

The crude extracts from eleven *S*. *lunaelactis* strains grown on the ISP7 solid medium were collected and subjected to UPLC-HRMS/MS analysis. As explained above, metabolite profiling was performed to search for molecules with *m/z* comprised between 700 and 800 Da that were only produced by the two selected *S*. *lunaelactis* strains that possess pSLUN1a (MM109^T^ and MM37), and not by the nine other selected strains that instead have pSLUN1b (MM22, MM25, MM28, MM31, MM40, MM78, MM83, MM113, and MM115) (Figure 2b). As shown in the heatmap of Figure 2b, a series of *m/z* signals displayed contrasting production patterns between the two groups of *S*. *lunaelactis* strains. An analysis of the extracted ion chromatograms (EIC) of the full extract of the 11 *S*. *lunaelactis* strains revealed a series of peaks only present in strains harboring pSLUN1a (Figure 2c). The most intense signal has an *m*/*z* of 725.4092, which is appropriate for the molecular formula of C_35_H_53_N_10_O_7_^+^ (0.17 ppm error) (Figure 2d). In the same HRMS spectrum, a second ion with lower signal intensity corresponds to the [M + Na]^+^ ion at *m*/*z* 747.3903, which corresponds to the molecular formula C_35_H_52_N_10_O_7_Na^+^. The masses and formula of both proton and sodium adducts reveal that the main compound of BGC 28a has a molecular formula of C_35_H_52_N_10_O_7_ (monoisotopic mass of 724.4020 Da), which from now on will be referred to as compound **1** (hereafter named lunaemycin A).

Interestingly, the experimentally calculated mass of compound **1** only differs by 16 Da compared to the mass of the in silico predicted cyclized structure of the main product of BGC 28a (Figure 1). From the molecular formula C_35_H_53_N_10_O_7_ inferred from *m*/*z* 725.41, and according to the amino acid building blocks (D-*allo*-Ile, L-Phe, D-Piz, L-Piz, D-Piz, L-Piz) predicted to be loaded by adenylation domains, compound **1** could correspond to either i) an open peptide with an extra unsaturation from a dehydrogenation step or ii) a cyclic peptide carrying a hydroxyl group added in an oxidation step.

An analysis of the HRMS/MS spectrum of the *m*/*z* 725.41 molecular ion revealed sequence tags that allowed us to propose that compound **1** is the cyclized sequence of the hexapeptide Phe-OH-Ile-Piz-Piz-Piz-Piz (Figure 3). Indeed, the daughter ion fragments of *m*/*z* 596.33, 449.26, 337.2, 225.13, and 113.07 correspond to consecutive losses of (i) an hydroxy-leucine (H_2_O neutral loss-derived tag fragment of *m*/*z* 129.08), (ii) a phenylalanine (H_2_O neutral loss-derived tag fragment of *m*/*z* 147.07), followed by three consecutive losses of Piz residues (three H_2_O neutral loss-derived tag fragments of *m*/*z* 112.06), and finally, the ion fragment tag of *m*/*z* 113.07 corresponding to Piz+H^+^ (Figure 3b). The fragmentation spectrum thus confirmed the presence of tags associated with all six amino acids predicted to be incorporated by the A domains of the active modules of BGC 28a. The only difference with the initial prediction is the hydroxylation of the Ile residue, which could be attributed to the product of *lun21* (SLUN_RS38510, Table 1) encoding a FAD-dependent oxidoreductase, homologous to the oxidoreductase PlyE of *Streptomyces* sp. MK498-98 F14 involved in the production of polyoxypeptins [40]. Indeed, PlyE is predicted to be responsible for the N-hydroxylation of valine and alanine building blocks on the nitrogen atom between Piz and Val or Ala residues of polyoxypeptin A. A similar role is also proposed for the FAD-dependent monooxygenase CchB that catalyzes the N-hydroxylation of the δ-amino group of ornithine in coelichelin biosynthesis before its recruitment by the NRPS machinery [41]. Similarly, Lun21 would perform the N-hydroxylation of the Ile residue, which would lead to the structure proposed for compound **1** (Figure 3b). Whether this N-hydroxylation of the Ile building block is performed (i) before its recruitment by the A domain of Lun20 (as suggested for coelichelin biosynthesis), (ii) once it is activated and tethered on the thiolation (T) domain of Lun20 (as proposed for polyoxypeptin synthesis) or after the formation and cyclization of the hexapeptide remains speculative at this stage.

To strengthen the reliability of the proposed structure of compound **1**, the *m*/*z* of the MS/MS data were correlated to the structure of ions produced during the fragmentation mechanism (Figure 4a). The *m*/*z* of nine fragment ions can only correlate with the structure of ions containing one to four Piz residues (fragments 1–9 in blue in Figure 4b). Four of these fragment ions—*m*/*z* 113.0711, 225.1347, 337.1984, and 449.2608—can be linked to structures containing respectively 1, 2, 3, and 4 Piz residues, expected to result from sequential *b_x_-y_z_* fragmentation steps. Additionally, *m*/*z* 309.2034 and 197.1394 result from the neutral loss of CO of the aforementioned ions *m*/*z* 337.1984 and 225.1347, respectively. At least seven fragment ions contain phenylalanine’s aromatic side chain thereby confirming the presence of this amino acid in the structure of compound **1** (fragments 10–16 in green in Figure 4c). Fragment ions with *m*/*z* 260.1396, 372.2032, 484.2668, and 596.3304 arise from sequential *b_x_-y_z_* fragmentation steps and represent phenylalanine connected to increasing numbers of piperazate residues, whereas the remaining ions (*m*/*z* 232.1445, 456.2719, and 568.3356) are associated with a subsequent CO loss step.

It is interesting to note that no hydroxylation can be inferred from the seventeen fragments discussed above, confirming that the hydroxylation in compound **1** must reside in the remaining amino acid, i.e., isoleucine. This is confirmed by the presence of four fragments that contain a hydroxyl-isoleucine residue (*m*/*z* 214.1551, 242.1500, 347.1841, and 354.21366, fragments 17–20 in red in Figure 4d), all of which resulted from similar fragmentation steps, as discussed for the other ions (sequential *b_x_-y_z_* fragmentation and CO loss). Finally, the hydroxylation of compound **1** and the hypothesis of a hydroxylated isoleucine in its structure are further confirmed by the presence of four fragment ions seemingly resulting from the neutral loss of H_2_O (fragments 21–24 in orange in Figure 4e). The fragment ion with *m*/*z* 707.3989 is the result of water loss from the [M+H]^+^ adduct and can generate the ion itself with *m*/*z* 679.4040 from CO loss. The two other fragment ions (*m*/*z* 196.1447 and 224.1394) would come from the loss of water from the previously mentioned isoleucine-bearing fragments *m*/*z* 214.1551 and 242.1500, respectively.

#### 2.2.2. Structure Elucidation by NMR Spectroscopy

Finally, nuclear magnetic resonance (NMR) was performed in order to confirm the predicted and MS/MS-deduced structure of compound **1**. Indeed, even though we can confidently propose the presence of a hydroxy-leucine residue in the structure of compound **1**, the MS/MS information alone does not allow to specify the position of the hydroxyl group in this residue. The fragment structures proposed in Figure 4d assume a hydroxylation on the amide nitrogen of isoleucine, which can only be confirmed by NMR. Multiple media were tested (OSMAC approach [42] and known elicitors of secondary metabolite production in *Streptomyces* spp. [43,44,45]) to assess the optimal culture conditions for compound **1** production in order to obtain sufficient material for NMR analysis. An analysis of the metabolite profiles showed that the solid ISP1 medium supplemented with N-acetyl-D-glucosamine 25 mM led to the best production yield of compound **1** (not shown). The dried ethyl-acetate full extract of *S*. *lunaelactis* MM109^T^ was processed using the ÄKTA liquid chromatography purification system, followed by semi-preparative HPLC (SP-HPLC) purification, which resulted in 2.15 mg of a pale-yellow-white powder from ~0.5 L of solid culture. UPLC-HRMS/MS revealed that this fraction contained two compounds with similar molecular weights, the most intense molecular ion peak being compound **1** (*m*/*z* of 725.4092), and a second molecular ion peak with an *m*/*z* of 723.3935 (compound **2**). This *m*/*z* value suggests that compound **2** is a dehydrogenated derivative of compound **1** that possibly contains a dehydro-Piz moiety, as was described previously for antrimycins [46]. This hypothesis was confirmed by ^1^H NMR results (see below) and by HRMS/MS analysis (see the following section on the structural diversity of lunaemycins).

The semi-purified powder containing compound **1** and minor amount of compound **2** was submitted to ¹H NMR (700MHz), ¹³C NMR (176 MHz) as well as 2D NMR (COSY, HMBC, and HSQC). 1D- and 2D NMR data were analyzed with MestReNova V.14 and allowed to confirm the chemical structure predicted by the in silico analysis of BGC 28a and deduced from the MS/MS fragmentation pattern. Table 2 shows the structure and NMR assignment proposed for compounds **1** and **2** (Figure 5a,b and Appendix A); however, the overlapping signals did not allow a complete structural assignment, as multiplicities and integrations were not unambiguously characterized.

Considering the chemical shifts observed in the ^1^H and ^13^C NMR spectra, signature signals typical of cyclopeptides were observed, revealing peptide bonds between amino acid residues with no amino or carboxy interruption of the amino acid sequence. ^1^H NMR data revealed the presence of the six α hydrogens of the hexapeptide core between δ 4.8 and δ 5.8 that can be attributed to compound **1** (δ 5.82–5.76 for the Ile; δ 5.36–5.28 for the Phe; δ 4.92–4.84, 5.39, 5.45, and 5.75 for the Piz residues), and compound **2** (δ 5.96 for the Ile; δ 5.68 for the Phe; δ 4.92–4.84, 5.01, 5.36–5.28, and 5.67 for the Piz residues). Further confirmation comes from the ^13^C-NMR signals between δ 170.3 and δ 174.1, attributed to the peptide carbonyl groups (C-6, C-13, C-19, C-25, C-31, C-41), and between δ 46.7 and δ 56.4, attributed to the α carbons (C-2, C-12, C-18, C-24, C-30, C-33).

Based on previously published data [47], a series of multiplets between δ 1.2 and δ 2.3 in the ^1^H NMR spectra can be attributed the eight β- and γ-methylene groups from the piperazate residues present in the structure of compound **1** (C-10, C-11, C-16, C-17, C-22, C-23, C-28, and C-29). Additionally, a series of ^13^C NMR signals between δ 18.4 and δ 26.2 can be attributed to those same methylene groups. Similarly, the four δ-methylene groups (C-9, C-15, C-21, and C-27) can be correlated to the ^1^H NMR signals between δ 2.9 and δ 3.1 and to the ^13^C NMR signals between δ 46.9 and δ 47.3. Interestingly, a signal at δ 6.91 (d, *J* = 3.7 Hz, ^1^H for compound **2**) in the ^1^H NMR spectra, which is correlated on the HSQC to the ^13^C NMR signal at δ 142.9, can be attributed to a methylene hydrogen of an imine group. This observation supports the hypothesis of a replacement of a piperazate residue by a dehydro-piperazate in compound **2**.

The presence of phenylalanine and isoleucine residues in the structure of compounds **1** and **2** is clearly detectable from the results of NMR experiments (Table 2). The six aromatic carbons (C-35 to C-40) of the phenylalanine’s benzene ring were observed at δ 137.7 (quaternary carbon C-35); δ 128.4 (two meta carbons, C-37 and C-39); 129.7, (two ortho carbons, C-36 and C-40); and δ 126.6 (the para carbon C-38). Also, the five aromatic hydrogens corresponding to this benzene ring were observed in ^1^H NMR spectra between δ 7.1 and δ 7.2. The two hydrogens at meta position (linked to C-37 and C-39) were attributed to the signal at δ 7.24 and could be discerned from the three hydrogens on ortho (linked to C-36 and C-40) and para (linked to C-38), seen at δ 7.2–7.14. As for the non-aromatic carbons in those residues, we were able to attribute ^13^C signals to the two methylenes (δ 25.2 for C-4 from the Ile and δ 37.7 for C-34 from the Phe), to the tertiary carbon C-3 from the Ile (δ 32.8), and to the two methyls, C-5 and C-7, from the Ile (δ 11.1 and 15.5, respectively). The upfield region of the ^1^H NMR spectrum shows signals corresponding to the non-aromatic hydrogens of the minor (**2**) and major (**1**) compounds of the sample analyzed by NMR. The methylene hydrogen of isoleucine, linked to C-4, can be attributed to δ 1.02–0.94, while the methylene hydrogens of phenylalanine, linked to C-34, were attributed to δ 2.93–2.86 and 2.77–2.67. For the isoleucine, the hydrogen on C-3 was attributed to peaks between δ 2.12–2.03, and the methyl hydrogens linked to C-5 and C-7 were attributed to δ 0.82–0.76 and δ 0.84, respectively.

The observation of the non-substituted s-butyl group of isoleucine indicates that hydroxylation of **1** is not localized at its side chain, suggesting that the hydroxylation should take place at the amide nitrogen. Additionally, 2D NMR COSY and ^1^H-^13^C HMBC correlations, shown in Figure 5c and highlighting proton-proton and proton-carbon correlations respectively, further supported this conclusion. Another key observation is the presence of a single HMBC correlation between an alpha NH ^1^H NMR signal at δ 8.27 (attributed to phenylalanine’s NH) and an alpha carbonyl ^13^C NMR signal at δ 172.7 (attributed to the neighboring piperazate carbonyl). Those observations allowed us to further confirm the N-hydroxylated structure previously proposed for Lunaemycin A.

In conclusion, combined in silico, HRMS/MS, and NMR analyses demonstrated that the main product of BGC 28a, compound **1**, is a cyclic hexapeptide of a mass of 724.4019 Da and the molecular formula C_35_H_52_N_10_O_7_, corresponding to the cyclized amino acid sequence D-Phe-OH-L-Ile-D-Piz-L-Piz-D-Piz-L-Piz. We have interrogated StreptomeDB 3.0 [48] and The Natural Products Atlas 2.0 [49] databases and found no natural compounds with neither identical molecular formula, monoisotopic mass, nor molecular weight. Interrogation of databases with broader chemical compounds (NIST, Pubchem, and Chemspider) identified ‘SCHEMBL12568990′ (Substance SID: 237623651; Compound CID: 53234730 in the Pubchem database) retrieved from patent US-2011136752-A1 [50], as the closest match of compound **1**, sharing the same mass, same molecular formula, and same proposed structure. We decided to call compound **1** lunaemycin A, with the Latin prefix lunae- (L. gen. n. lunae, of the moon) referring to the moonmilk speleothems where *S*. *lunaelactis* MM109^T^ was originally isolated and the suffix -mycin, which refers to antibiotic compounds derived from bacteria with fungus-like structure such as *Streptomyces*. The antibiotic activity of lunaemycins A and lunaemycin-related compounds will be described in Section 2.4.

### 2.3. Structural Diversity of Lunaemycins

The comparative metabolomic study of metabolites with masses ranging between 700 and 800 Da (Figure 2b) suggested a broad diversity of lunaemycin-derived molecules produced by *S*. *lunaelactis* strains containing pSLUN1a. Using the MS/MS data, we explored this diversity via global natural products social molecular networking (GNPS) as performed previously [22]. One constellation comprises 42 nodes associated with compounds only produced by the two selected *S*. *lunaelactis* strains that possess BGC 28a (MM37 and MM109^T^) and includes the ion at *m*/*z* 725 corresponding lunaemycin A (Figure 6).

Figure 7 presents the detection of the most abundant compounds identified as lunaemycin derivatives produced by *S*. *lunaelactis* MM109^T^, as well as their HRMS spectra and their deduced molecular formula. Comparison of the MS/MS spectra allowed us to identify the position of the chemical modification(s) for 14 of these molecules compared to lunaemycin A (Appendix A, Table 3), whereas other five lunaemycin derivatives were produced in too weak amounts for proper analysis. Table 3 lists all compounds clustering in the constellations of lunaemycins, their proposed names, retention times, *m*/*z* signals, and their molecular formulas.

The HRMS/MS-based identification of the amino acid building-blocks composing the 15 lunaemycins is detailed in Appendix A. Eight lunaemycins possess at least one dehydrogenated-Piz (dh-Piz, corresponding to the neutral loss of 110.5) instead of a classical Piz. More precisely, if we consider the linear sequence of Lunaemycin A as Hydroxy-Ile-Piz_1_-Piz_2_-Piz_3_-Piz_4_-Phe, Piz_1_ is replaced by a dh-Piz in lunaemycin E1 (**5a**); Piz_2_ is replaced by a dh-Piz in lunaemycins B1 (**2a**), C1 (**3a**), and J1 (**10a**); Piz_3_ is replaced by a dh-Piz in lunaemycins B1(**2a**), C1 (**3a**), C2 (**3b**), E1 (**5a**), and J3 (**10c**); and finally, Piz_4_ is replaced by a dh-Piz in lunaemycin C1 (**3a**). Only lunaemycin F (**6**) presents the hydroxy-Piz residue (HO-Piz, corresponding to the neutral loss of 128.09) at the position of Piz_3_. Five lunaemycins present the hydroxydehydro-Piz residue (HO-dh-Piz, corresponding to the neutral loss of 126.05) at the position of Piz_4_ (lunaemycins D (**4**), E1 (**5a**), E2 (**5b**), G (**7**), and H (**8**). Finally, the HO-Ile is replaced by a dihydroxy-Ile (DIHO-Ile, corresponding to the neutral loss of 145.08) in lunaemycins G (**7**) and H (**8**), by an hydroxyl-Valine (HO-Val, corresponding to the neutral loss of 115.06) in lunaemycins I (**9**), J2 (**10b**), and J3 (**10c**), and by an Ile (corresponding to the neutral loss of 113.08) in lunaemycin J1 (**10a**).

### 2.4. Antibacterial Activity of Lunaemycins

Molecules produced by known BGCs closest to the *lun* cluster (himastatins, stenothricins, kutznerides, and aurantimycins) possess antimicrobial activities suggesting that lunaemycins could possibly display similar biological activities. Moreover, amongst the genes of the *lun* BGC, BLAST analyses also suggest that lunaemycins are likely to possess antibacterial and/or anti-proliferative activities. For instance, *lun9* and *lun10* encode for an efflux pump and its associated TetR-family transcriptional regulator, respectively, and whose homologues in other BGCs play a role in self-resistance of the producing organism. In addition, *lun25* and *lun26* and their closest homologues in the BGCs of aurantimycins (ArtJ and ArtK) and the polyoxipeptins (PlyJ and PlyK), encode for an ABC transporter system belonging to the DrrA-DrrB family also conferring self-resistance to the producers [51].

Antibacterial activity of lunaemycins A, B1, and D (0.1 mg/mL, ~13.5–14 mM) was first performed via agar-diffusion assays which showed significant direct-acting activity on all tested Gram-positive bacteria except on *Enterococcus hirae* (though a small zone of growth inhibition is observed at the periphery of the disc suggesting more susceptibility to lunaemycins compared to gentamicin 0.1 mg/mL used as positive control) (Figure 8). In contrast, no growth inhibition was reported on Gram-negative bacteria (Figure 8). Overall, similar antibacterial activities could be observed for lunaemycins A and B1, whereas lunaemycin D showed weaker growth inhibiting activities.

Minimal inhibitory concentration (MIC) and minimal bactericidal concentration (MBC) values were determined with pure lunaemycin A. The potent direct-acting activity observed in agar diffusion assays was confirmed, with MIC values ranging 0.12–0.25 µg/mL on type strains and, more interestingly, also on clinical isolates showing various resistance phenotypes (Table 4). The MBC values were comprised between 0.25 and 8 µg/mL. Interestingly, the MBC values obtained with some antibiotic resistant clinical isolates, especially those showing resistance to methicillin, vancomycin, or linezolid, were overall lower (from 0.25 to 2 µg/mL) than that of other strains, indicating a potent bactericidal activity of luneamycin A for these strains. As largely reported in the literature (e. g., references [52,53]), some of these resistance mechanisms are associated with significant biological or fitness cost, potentially explaining the lower MBC values measured with these strains. Expectedly, no activity could be observed on both susceptible and resistant Gram-negative isolates, as previously observed via the agar-diffusion assays.

## 3. Materials and Methods

### 3.1. Bacterial Strains and Culture Conditions

All bacterial strains used in this study are listed in Table 5. The archetype strain *Streptomyces lunaelactis* MM109^T^ [20,21] and all other *S*. *lunaelactis* strains used in this study were isolated from moonmilk deposits of the cave ‘la grotte des collemboles’ (Comblain-au-Pont, Belgium) [9,10]. *Streptomyces* spores and mycelium stocks were prepared as described in [54]. Cultivation media were prepared as described in [54], and inoculated plates were incubated for a week at 28 °C.

### 3.2. Bioinformatics and In Silico Analyzes

Mining for BGC in pSLUN1a (NZ_CP026305.1) and identification of closest homologous genes in known BGCs in MiBIG database [27] were performed with antiSMASH [55]. Sequence of linear plasmid pSLUN1b can be accessed from (https://www.ncbi.nlm.nih.gov/search/all/?term=JABSUF010000009, accessed on 17 June 2020). Prediction of amino acid building blocks recruited by adenylation domains was performed with SANDPUMA [39] and NRPSpredictor2 [56]. Visualization of the molecular network (metabolite diversity and relatedness)) was performed via the Global Natural Products Social Molecular Networking website (GNPS, https://gnps.ucsd.edu/, accessed on 17 June 2020) from the tandem mass (MS/MS) spectrometry data obtained from the culture extracts od the different S. *lunaelactis* strains as previously described [22]. 1D- and 2D NMR data were analyzed with MestreNova V.14 (https://mestrelab.com/download_file/mnova-14-0-0/, accessed on 17 June 2020).

### 3.3. Compound Identification by Ultra-Performance Liquid Chromatography–Tandem Mass Spectrometry (UPLC–MS/MS)

Solid media inoculated with *S*. *lunaelactis* were cut into small pieces and mixed overnight in agitator with an equal volume of ethyl acetate to extract metabolites. After centrifugation at 4000 rpm for 20 min, the supernatant was evaporated to dryness by a rotavapor^®^ (IKA RV10 digital, VWR, Radnor, PA, USA) at 25 °C under 210 rpm and suspended in 2 mL of acetonitrile. Extracts were analyzed by Ultra-Performance Liquid Chromatography-High Resolution Mass Spectrometry (UPLC–HRMS) and by Ultra-Performance Liquid Chromatography–Tandem Mass Spectrometry (UPLC–MS/MS). Lunaemycins were identified according to their exact mass, the isotopic pattern, their MS/MS spectrum of the molecular ion HCD fragmentation, and the UV-VIS absorbance spectra. The detailed protocols for lunaemycins extraction and purification are detailed in Point 3.4. and Appendix B, Appendix B, Appendix B, Appendix B and Appendix B

### 3.4. Nuclear Magnetic Resonance (NMR)

The production and purification of lunaemycins was performed as follows: the ethyl acetate extract of *S*. *lunaelactis* MM109^T^ grown on 25 agar plates of ISP1 + GlcNAc (~500 mL volume) was dried with rotary evaporator (Rotavapor^®^ R-100) which allowed to obtain 126 mg of a pale yellow powder, re-dissolved in 60 mL of an acetonitrile/water (MilliQ filtered) (10:3, *v*/*v*) solution, and fractioned on a C18 LC column (Phenomenex) using an ÄKTA™ purification system. Metabolite elution was performed by using a gradient of increasing concentration of acetonitrile (LC method, Appendix B), and each of the 90 fractions were analyzed by HPLC (HPLC method, Appendix C) to detect peaks corresponding to lunaemycins. Fractions containing the compounds of interest (fractions 35, 36, and 37 to 41) were subjected to a second round of purification using semi-preparative HPLC (SP-HPLC method, Appendix D). Fractions 37 to 41 were pooled, and the volume of each fraction was reduced to 2 mL by evaporation (under gas flow), which served for twenty semi-preparative HPLC injections with 100 µL of each fraction. From fraction 35, we generated 21 sub-fractions among which 4 of them contain lunaemycin D (Table 3) as subsequent UPLC-HRMS analysis (UPLC Method Appendix E, and the MS Method, Appendix F) identified the molecular ion species of *m*/*z* 739.39 [M+H]^+^. Fraction 36 generated 19 sub-fractions, among which sub-fractions 3 to 5 contain lunaemycin D (previously identified in fraction 35), and sub-fractions 17 to 19 contain lunaemycin B1, as confirmed by UPLC-HRMS analysis that detected the molecular ion species of *m*/*z* 723.40 [M+H]^+^. Finally, the pooled fractions 37–41 generated 40 sub-fractions (SP-HPLC method, Appendix D), among which sub-fractions 23 to 28 mainly contain lunaemycin A (ion species of *m*/*z* 725.41 [M+H]^+^, see Figure 4a), and lower amounts of lunaemycins B1/B2 (ion species of *m*/*z* 723.40 [M+H]^+^, see Figure 4a). The fractions containing a single chromatographic peak were pooled and evaporated under continuous gas flow and resuspended in 1 mL of water/acetonitrile (1/9, *v*/*v*) solution and diluted to reach a concentration of 0.1 mg/mL for subsequent antibacterial assays (see point 4.5 below).

The pure dried extract mainly containing lunaemycins A (compound 1) was dissolved in 250 µL of hexadeuterodimethylsulfoxide (DMSO-d_6_) containing 1% (*v*/*v*) of tetramethylsilane (TMS) as a reference (Sigma-Aldrich, St. Louis, MO, USA) and transferred in a 3 mm NMR tube (103.5 mm long, Bruker). NMR spectra were acquired using a Bruker Avance NEO 700 MHz spectrometer equipped with a cryoprobe and Bruker’s TopSpin 3.5 software package, in order to perform ^1^H, ^13^C (APT), COSY, ^1^H-^13^C HMBC, and ^1^H-^13^C HSQC experiments (standard Bruker parameters), and data were interpretated using the MestreNova V.14 software. Sample temperatures were controlled with the variable-temperature unit of the instrument.

### 3.5. Antibacterial Assays

Pure lunaemycins A, B1, and D were obtained as described above in point 4.4. Agar-diffusion tests were implemented as described previously [24]. Briefly, few microliters of an overnight culture (10 µL of the bacterial stock spread on the solid LB or BH media) were used to make bacterial suspensions with standardized turbidity (OD600 between 0.08 and 0.1, corresponding to a 0.5 McFarland turbidity standard). A total of 100 µL of the standardized bacterial suspension were then inoculated on solid LB or BH media, and 3 mm thick Whatman^®^ papers discs (Sigma-Aldrich, St. Louis, MO, USA) containing 10 µL (2 × 5 µL) of antibiotics are placed on the freshly inoculated solid medium. After overnight incubation at 37 °C, the antibacterial activity was evaluated by measuring the diameter of growth inhibition. Control conditions were performed using 100 µg/mL gentamicin as a positive control (corresponding to 1 µg of gentamicin placed on the positive control disc), and acetonitrile, DMSO or water as a negative control. Additional antibacterial susceptiblity assays were performed at the High-Throughput Screening and Protein Engineering platform (HiProtEn, Dipartimento di Biotecnologie Mediche—Università di Siena). Minimal inhibitory concentration (MIC) and minimal bactericidal concentration (MBC) values were determined in Mueller-Hinton broth, as recommended by CLSI (document M07-A10, 2015 [57]) and the European Committee on Antimicrobial Susceptibility Testing (EUCAST document “Terminology relating to methods for the determination of susceptibility of bacteria to antimicrobial agents”, 2000 [58]) on reference (type) strains and clinical isolates as indicated (Table 5).

## 4. Conclusions

Herein, we report the discovery and structural elucidation of lunaemycins: new, cyclic hexapeptide antibiotics only produced by *S*. *lunaelactis* species that possess the linear plasmid pSLUN1a. The potent antibacterial activity of lunaemycins against antibiotic resistant Gram-positive bacteria may in part explain how *S*. *lunaelactis* strains have found their own place in extremely oligotrophic and competitive environments such as moonmilk deposits. Next to the proposed structure of 15 lunaemycin derivatives, we also propose the biosynthetic pathway for lunaemycin A production, the main compound of the *lun* BGC of *S*. *lunaelactis* MM109^T^ (Figure 9).

Our work provides a good example in which a deep in silico analysis of genes that compose a BGC combined with an exhaustive literature overview of homologous genes allowed us to maximize the accuracy of the predicted structure of a cryptic compound. This in silico approach highly facilitated the comparative mass spectrometry analysis by guiding the examination on a limited series of ions that possessed the appropriate fragmentation tags. Next to core biosynthetic genes, the *lun* BGC also provides an interesting case where genes required for the synthesis of the building block piperazic acid from ornithine are also present within the BGC itself, and even also the genes that convert the glutamate and glutamine precursors to ornithine. Such a situation was previously observed, notably in the BGC for quinichelin biosynthesis [59], and in many NRPS BGCs that necessitate Piz as building block [37]. This inclusion of genes/enzymes of metabolic pathways of primary metabolism within a BGC is most likely mandatory to provide sufficient building blocks at the proper timing of specialized metabolite production.

Another main take-home message of our work relates to the use of multiple strains of the same species to facilitate the correlation between the genetic material associated with the biosynthesis of natural compounds. The current tendency in strain prioritization is instead to try to exclude from a collection the multiple strains that belong to the same species in order to avoid the identification of the same bioactive molecules [22]. Here, we showed how strain redundancy instead facilitated our work, even avoiding the necessity to generate null mutants to link a BGC to its natural product. This is particularly interesting when a BGC is present on a linear plasmid whose presence in multiple copies complicates gene inactivation (about three copies for pSLUN1a in *S*. *lunaelactis* MM109^T^ [21]). Our attempts to delete genes of BGC 28a have been unsuccessful so far. We are currently applying the same strategy to identify and assess the biological activity of the cryptic products of BGC 28b found in the linear plasmid pSLUN1b only present in the second subgroup of *S*. *lunaelactis* strains [22].

## Figures and Tables

**Figure 1 ijms-24-01114-f001:**
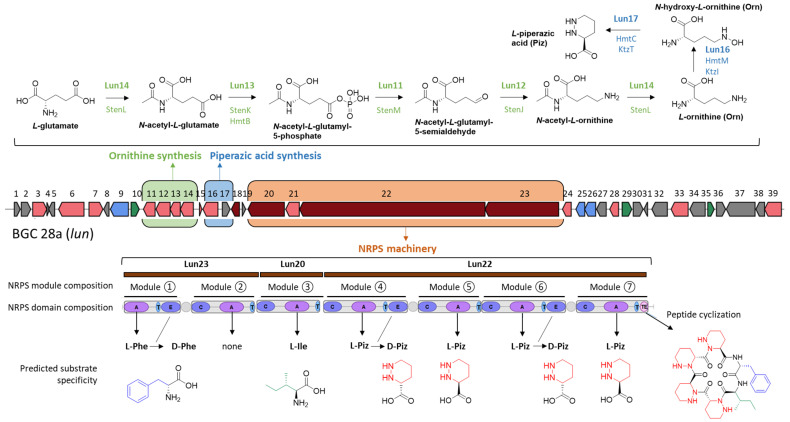
Predicted biosynthetic pathways and substrate specificity of BGC 28a (*lun* cluster). Top: genes/proteins predicted to be involved in the Ornithine (Orn) and Piperazic acid (Piz) biosynthesis. Bottom: modules, domains, predicted amino acid building-blocks, and modifying reactions of NRPSs of BGC 28a predicted to generate a cyclized hexapeptide with a monoisotopic mass of 708.4041 Da. Abbreviations: A, adenylation domain; C, condensation domain; T, thiolation domain; TE, thioesterase domain; Ile, isoleucine; Phe, phenylalanine; Piz, piperazic acid.

**Figure 2 ijms-24-01114-f002:**
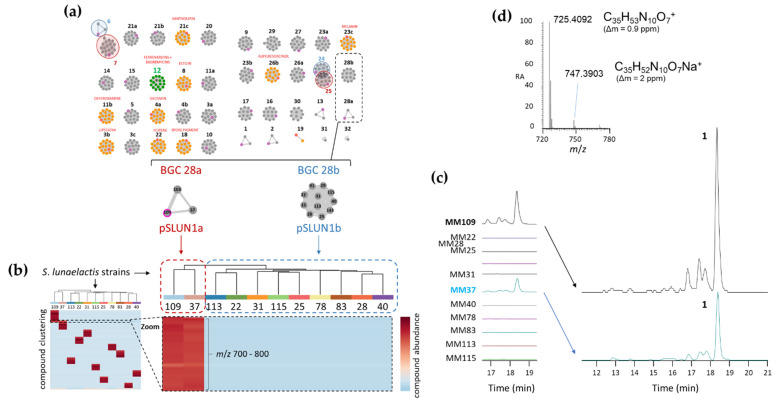
(**a**) Genome mining of *S*. *lunaelactis* strains where the 42 predicted BGCs are grouped in ‘grapes’ of nodes where each node represents a BGC in one strain (from [22]). Note the two grapes corresponding to BGC 28a and to BGC 28b found in strains either containing the linear plasmid pSLUN1a or pSLUN1b, respectively. (**b**) Heatmap of the metabolomic analysis of 11 *S*. *lunaelactis* strains grown on ISP7 media. The zoom on the heatmap highlights the differential production patterns for metabolites with *m*/*z* comprised between 700 and 800 Da and only present in the extract of *S*. *lunaelactis* strains that possess BGC 28a (MM109^T^ and MM37). (**c**) Extracted ion chromatograms (EIC, from *m*/*z* 700 to 800) of the full extract of the 11 strains with a focus on the retention times of signals associated with compounds only detected in *S*. *lunaelactis* strains with BGC 28a (MM109^T^ and MM37). (**d**) HRMS-predicted molecular formula of compound **1** (for both the proton and the sodium adducts) associated with the main signal detected in EICs of *S*. *lunaelactis* strains MM37 and MM109^T^.

**Figure 3 ijms-24-01114-f003:**
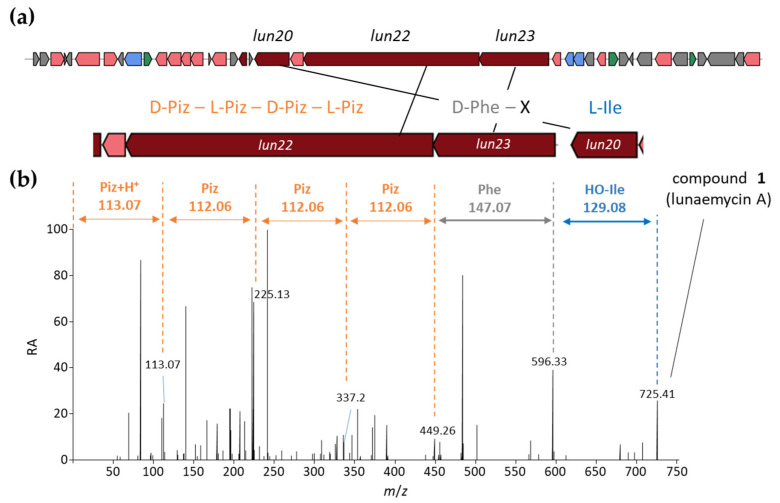
HRMS/MS-based identification of the amino acid building-blocks of compound **1**. (**a**) NRPS genes of BGC 28a (*lun* cluster) and the predicted amino acid building-block sequence. (**b**) MS fragmentation spectrum and proposed structure of compound **1** (lunaemycin A). Only fragments corresponding to amino-acid or peptide *m*/*z* of the predicted peptide sequence are highlighted. Color code: orange, piperazic acid (Piz); grey, phenylalanine (Phe); blue, hydroxylated isoleucine (HO-Ile).

**Figure 4 ijms-24-01114-f004:**
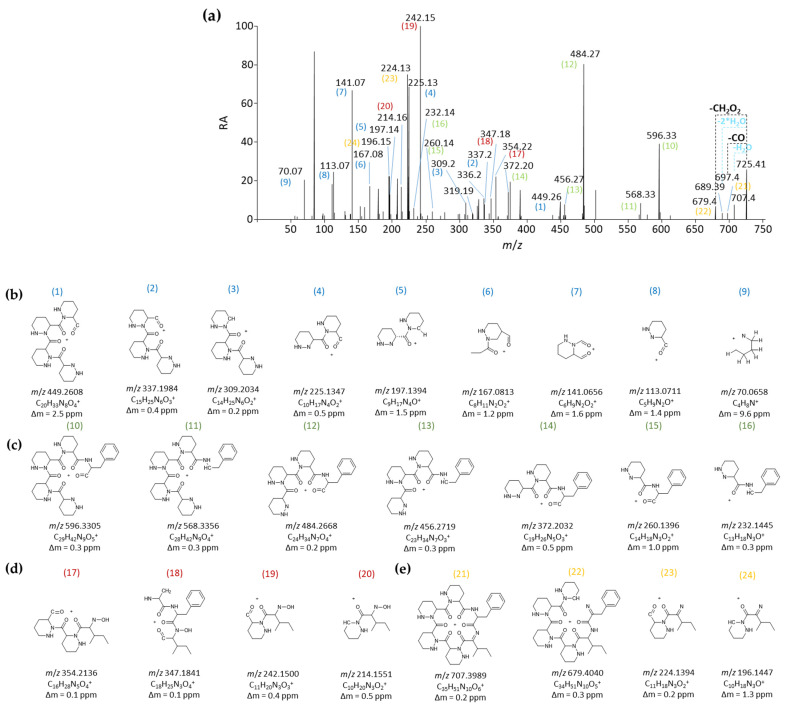
HRMS/MS-based identification of fragmentation patterns of compound **1**. (**a**) MS fragmentation spectrum. (**b**) Structure of ions containing one to four Piz residues (these ions are highlighted by numbers in blue in the MS fragmentation spectrum). (**c**) Structure of fragment ions containing the phenylalanine building block (these ions are highlighted by numbers in green in the MS fragmentation spectrum. (**d**) Structure of fragment ions that contain a hydroxyl-isoleucine residue (these ions are highlighted by numbers in red in the MS fragmentation spectrum). (**e**) Structure of fragment ions seemingly resulting from the neutral loss of H_2_O further supporting the presence of hydroxylated isoleucine (these ions are highlighted by numbers in orange in the MS fragmentation spectrum).

**Figure 5 ijms-24-01114-f005:**
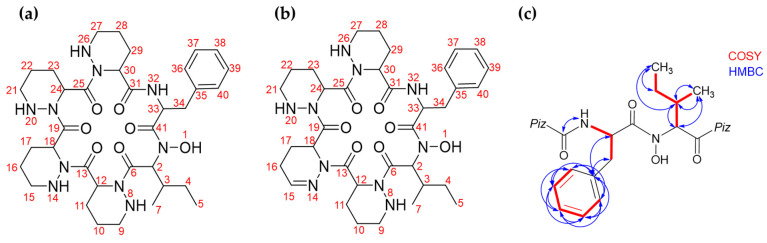
Structure elucidation of compounds **1** and **2** by NMR. (**a**) Compound **1**. (**b**) Compound **2**. (**c**) COSY and HMBC correlations that allowed to unambiguously confirm the location of the hydroxylation on compound **1**. See Appendix A.

**Figure 6 ijms-24-01114-f006:**
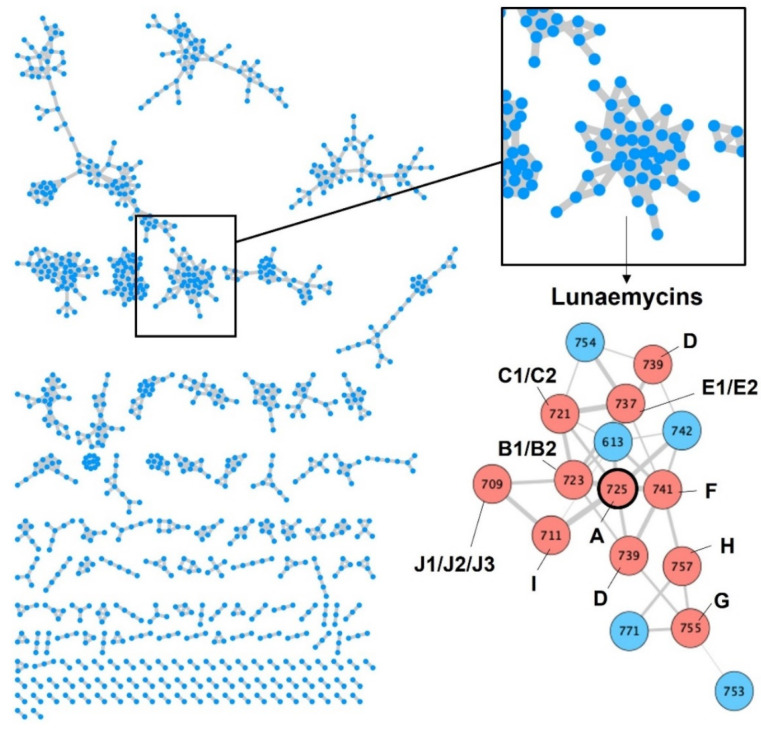
GNPS of metabolites extracted from *S. lunaelactis* strains. Left panel: Constellations of compounds constructed based on GNPS processing the UPLC-MS/MS data obtained from the full extract of the eleven *S*. *lunaelactis* strains grown on ISP7 medium. Each node in a constellation represents a parent mass (MS1) of a compound detected in one or more of the eleven full extracts analyzed. Right panel: Constellation of lunaemycins where nodes are linked by a straight line when they share similar fragmentation patterns. Pink and blue nodes: lunaemycins with solved and unsolved structure, respectively (see details in Table 3, Figure 7, and Appendix A).

**Figure 7 ijms-24-01114-f007:**
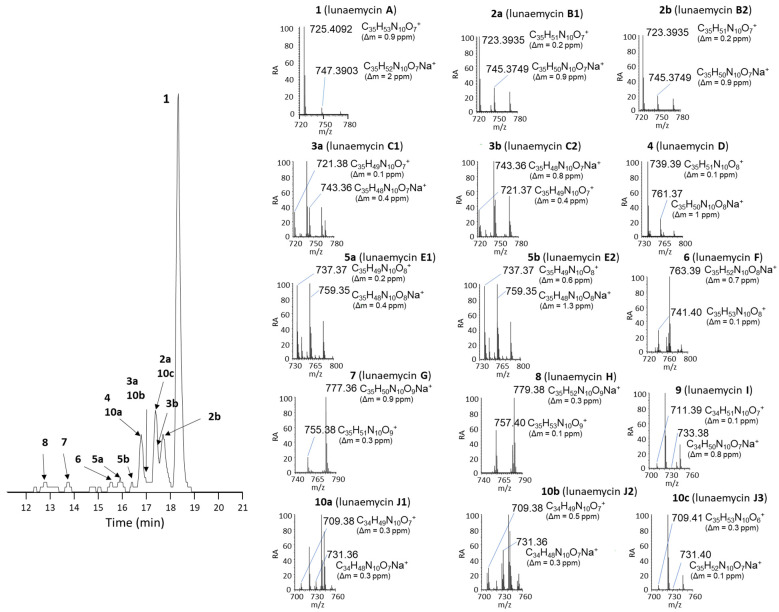
UPLC-HRMS chromatograms and mass spectra of lunaemycins. Left panel: Extracted ion chromatogram (EIC) with *m*/*z* signals ranging from 700 to 800 Da from the full extracts of *S*. *lunaelactis* MM109^T^ grown on ISP7 medium. Right panels: HRMS-predicted molecular formula of compound **1** to **10c** (for both the proton and the sodium adducts) associated with the main signal detected in the EIC displayed on the left panel.

**Figure 8 ijms-24-01114-f008:**
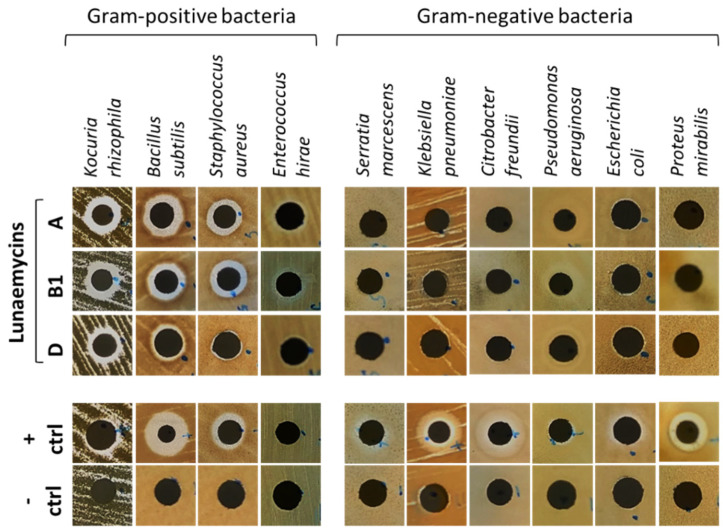
Agar-diffusion antibacterial activity assays of lunaemycins. 0.1 mg/mL of gentamicin was used as positive control, and a water/acetonitrile solution (1/9, *v*/*v*) was used as negative control.

**Figure 9 ijms-24-01114-f009:**
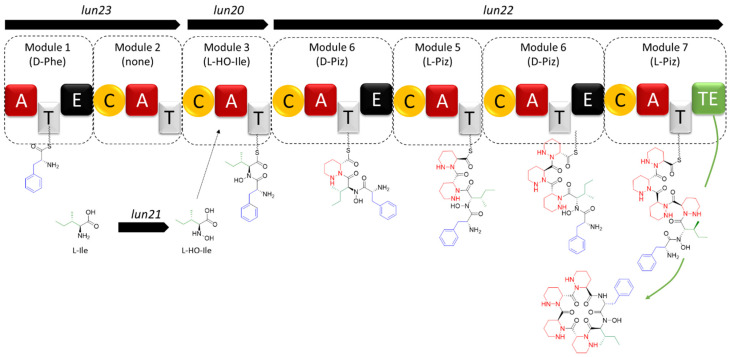
Proposed biosynthetic pathway of lunaemycin A. This model suggests that the Ile will be either N-hydroxylated before activation and recruitment by the A domain (like proposed for the coelichelin pathway) or hydroxylated once activated and tethered on the thiolation domain of Lun20 (like proposed for polyoxypeptin synthesis).

**Table 1 ijms-24-01114-t001:** Deduced functions of ORFs of the *lun* (BGC 28a) biosynthetic gene cluster of *S*. *lunaelactis* MM109^T^.

ORFs		Closest Homologue in MiBIG Database (Protein ID)	Homology (Id/Cov)	Proposed Function
Locus Tag Accession No.	Name Size (aa)			
SLUN_RS38410 WP_108155188.1	Lun1 177	TmcL, hypothetical protein, α,β-epoxyketone BGC of *Streptomyces chromofuscus* (CUX96959.1)	64/100	Unknown
SLUN_RS38415 WP_108155189.1	Lun2 278	DUF4097 family beta strand repeat-containing protein, α,β-epoxyketone BGC of *Streptomyces chromofuscus* (CUX96960.1)	48/100	Unknown
SLUN_RS38420 WP_108155190.1	Lun3 416	Cytochrome P450, streptovaricin BGC of *Streptomyces spectabilis* (ASZ00144.1)	61/97	Product modification
SLUN_RS38425 WP_108155191.1	Lun4 76	Ferredoxin, caniferolide A BGC from *Streptomyces caniferus* (QBF51761.1)	52/82	Product modification
SLUN_RS38430 WP_254708284.1	Lun5 179	GtrA family protein, monensin BGC from *Streptomyces cinnamonensis* (AAO65788.1)	60/82	Unknown
SLUN_RS38435 WP_159100457.1	Lun6 746	MPPL family transporter, alnumycin A BGC from *Streptomyces* sp. CM020 (ACI88887.1)	58/98	Exporter
SLUN_RS38440 WP_108155192.1	Lun7 409	MfnN, cytochrome P450, marformycin A BGC from *Streptomyces drozdowiczii* (AJV88386.1)	37/100	product modification
SLUN_RS38445 WP_108155193.1	Lun8 157	Hypothetical protein, no hit in MiBIG, closest to *Streptomyces* sp. Ag82_O1-15 (WP_095849664.1)	47/88	Unknown
SLUN_RS38450 WP_108155194.1	Lun9 531	MFS transporter, siamycin I BGC from *Streptomyces* sp. (BBD82029.1)	72/93	Resistance export
SLUN_RS38455 WP_108155195.1	Lun10 241	TetR/AcrR family transcriptional regulator, siamycin I BGC from *Streptomyces* sp. (BBD82030.1)	61/96	Resistance regulation
SLUN_RS38460 WP_108155196.1	Lun11 342	StenM, N-acetyl-gamma-glutamyl-phosphate reductase (ArgC), stenothricin BGC from *Streptomyces roseosporus* (EFE73305.1)	81/100	L-ornithine synthesis
SLUN_RS38465 WP_108155197.1	Lun12 417	StenJ, N(2)-acetyl-L-ornithine:2-oxoglutarate aminotransferase (ArgD), stenothricin BGC from *Streptomyces roseosporus* NRRL 15998 (EFE73302.1)	72/94	L-ornithine synthesis
SLUN_RS38470 WP_108155198.1	Lun13 300	StenK, N-acetylglutamate kinase (ArgB), stenothricin BGC from *Streptomyces roseosporus* (EFE73303.1)	81/97	L-ornithine synthesis
SLUN_RS38475 WP_108155199.1	Lun14 384	StenL, bifunctional N(2)-acetyl-L-ornithine:L-glutamate N-acetyltransferase (ArgJ), stenothricin BGC from *Streptomyces roseosporus* (EFE73304.1)	80/100	L-ornithine synthesis
SLUN_RS38480 WP_108155200.1	Lun15 70	KtzJ, aurantimycin A BGC from *Streptomyces aurantiacus* JA 4570 (WP_016638464.1)	71/100	Building block modification
SLUN_RS38485 WP_108155201.1	Lun16 444	HmtM, lysine N(6)-hydroxylase/L-ornithine N(5)-oxygenase, himastatin BGC of *Streptomyces himastatinicus* (CBZ42147.1)	63/97	Piz synthesis
SLUN_RS38490 WP_108155202.1	Lun17 230	HmtC, piperazate synthase, himastatin BGC *Streptomyces himastatinicus* (CBZ42137.1)	69/95	Piz synthesis
SLUN_RS38495 WP_108155203.1	Lun18 235	LmbU family transcriptional regulator, himastatin BGC *Streptomyces himastatinicus* (CBZ42138.1)	69/95	Regulation
SLUN_RS38500 WP_108155204.1	Lun19 100	chorismate mutase, no hit in MiBIG, *Streptomyces* sp. SLBN-118 (WP_142214144.1)	78/97	Phe biosynthetic pathway
SLUN_RS38505 WP_108155205.1	Lun20 1072	OciC, non-ribosomal peptide synthetase (Module 3, domain C-A-T), cyanopeptolin BGC from *Planktothrix agardhii* NIVA-CYA 116 (ABI26079.1)	43/98	Incorporated aa: L-Ile
SLUN_RS38510 WP_108155206.1	Lun21 400	NAD(P)/FAD-dependent oxidoreductase, aurantimycin A BGC from *Streptomyces aurantiacus* JA 4570 (WP_016638468.1)	53/99	Hydroxylation
SLUN_RS38515 WP_108155207.1	Lun22 5462	HmtL, non-ribosomal peptide synthetase (Modules 4/5/6/7, domains C-A-T-E/C-A-T/C-A-T-E/C-A-T-TE), himastatin BGC of *Streptomyces himastatinicus* (CBZ42146.1)	55–100	Incorporated aa: D-Piz/L-Piz/D-Piz/L-Piz and peptide release and cyclization
SLUN_RS38520 WP_159100458.1	Lun23 2167	Non-ribosomal peptide synthetase (Module 1/2, domains A-T-E/C-A-T), aurantimycin A BGC from *Streptomyces aurantiacus* JA 4570 (WP_016638470.1)	53/100	Incorporated aa: D-Phe/inactive
SLUN_RS38525 WP_108155209.1	Lun24 259	KtzF thioesterase, alpha/beta fold hydrolase, himastatin BGC of *Streptomyces himastatinicus* (CBZ42144.1)	58/97	Peptide release
SLUN_RS38530 WP_108155210.1	Lun25 258	ABC transporter permease, aurantimycin A BGC from *Streptomyces aurantiacus* JA 4570 (WP_016638474.1)	51/100	Drug resistance
SLUN_RS38535 WP_108155211.1	Lun26 323	Daunorubicin resistance protein DrrA family ABC transporter ATP-binding protein, aurantimycin A BGC of *Streptomyces aurantiacus* JA 4570 (WP_016638473.1)	66/94	Drug resistance
SLUN_RS38540 WP_108155212.1	Lun27 286	DUF4097 family beta strand repeat-containing protein, himastatin BGC *Streptomyces himastatinicus* (CBZ42150.1)	39/100	Unknown
SLUN_RS38545 WP_108155213.1	Lun28 261	3-oxoacyl-ACP reductase FabG, kiamycin biosynthetic gene cluster from *Streptomyces* sp. W007 (EHM27508.1)	74/97	Unknown
SLUN_RS38550 WP_108155214.1	Lun29 285	SARP family transcriptional regulator, mitomycin biosynthetic gene cluster from *Streptomyces lavendulae* (AAD28451.1)	30/90	Regulation
SLUN_RS38555 WP_108155215.1	Lun30 293	class I SAM-dependent methyltransferase, 5-isoprenylindole-3-carboxylate β-D-glycosyl ester BGC from *Streptomyces* sp. RM-5–8 (ANA09433.1)	31/85	Unknown
SLUN_RS38560 WP_108155216.1	Lun31 108	ArsR family transcriptional regulator, no hit in MiBIG database, *Streptomyces mangrovisoli* (WP_052743136.1)	93/85	Regulation
SLUN_RS38565 WP_108155217.1	Lun32 457	M14 family zinc carboxypeptidase, capreomycin IA BGC from *Saccharothrix mutabilis* subsp. Capreolus (ABR67765.1)	45/98	Unknown
SLUN_RS38570 WP_108155218.1	Lun33 500	Amino acid permease, amphotericin B biosynthetic gene cluster from *Streptomyces nodosus* (AAV48836.1)	54/89	Unkown
SLUN_RS38575 WP_257153969.1	Lun34 450	glutamine synthetase family protein, colabomycin E BGC from *Streptomyces aureus* (AIL50155.1)	36/100	Unknown
SLUN_RS38580 WP_108155220.1	Lun35 190	MarR family transcriptional regulator, no hit in MiBIG database, *Streptomyces* sp. ISL-44 (WP_215144797.1)	83/93	Regulation
SLUN_RS38585 WP_108155221.1	Lun36 268	acetoacetate decarboxylase family protein, enduracidin BGC from *Streptomyces fungicidicus* (ABD65946.1)	35/99	Unknown
SLUN_RS38590 WP_108155222.1	Lun37 846	SpoIIE family protein phosphatase, laidlomycin BGC from *Streptomyces* sp. CS684 (AFL48546.1)	48/68	Unknown
SLUN_RS38595 WP_108155223.1	Lun38 256	gamma-glutamyl-gamma-aminobutyrate hydrolase, nenestatin biosynthetic gene cluster from *Micromonospora echinospora* (ARD70860.1)	35/95	Unknown
SLUN_RS41955 WP_108155224.1	Lun39 495	Putative aldehyde dehydrogenase, pederin BGC from uncultured bacterium (AAS47555.1)	49/100	Unknown

**Table 2 ijms-24-01114-t002:** Assignment of NMR signals of the major (**1**) and minor (**2**) components of the semi purified sample.

Atom Number	Major Compound (1)		Minor Compound (2)	
	^1^H NMR *^a^*	^13^C NMR *^b^*	^1^H NMR *^a^*	^13^C NMR *^b^*
1	9.27 (b, 1H)		9.64 (b, 1H)	
2	5.82–5.76	56.4	5.96 (d, *J* = 11.0 Hz, 1H)	55.6
3	2.07–2.03	32.8	2.12–2.07	32.3
4	1.02–0.94	25.2	1.02–0.94	25.2
5	0.82–0.76	11.1	0.82–0.76	10.9
6		172.7		172.2
7	0.88–0.84	15.5	0.88–0.84	15.4
8	4.92–4.84 *^c^*		4.92–4.84 *^c^*	
9	2.9–3.1 *^d^*	46.9 *^d^*	* ^e^ *	* ^e^ *
10	1.2–2.3 *^f^*	18.4–26.2 *^f^*	* ^e^ *	* ^e^ *
11	1.2–2.3 *^f^*	18.4–26.2 *^f^*	* ^e^ *	* ^e^ *
12	5.75 (d, *J* = 5.6 Hz, 1H) *^g^*	48.9 *^g^*	5.82–5.76 *^g^*	47.6 *^g^*
13		173.5 *^h^*		171.3 *^h^*
14	5.05 (d, *J* = 13.8 Hz, 1H) *^c^*			
15	2.9–3.1 *^d^*	47.1 *^d^*	6.91 (d, *J* = 3.7 Hz, 1H) *^i^*	142.9 *^i^*
16	1.2–2.3 *^f^*	18.4–26.2 *^f^*	* ^e^ *	* ^e^ *
17	1.2–2.3 *^f^*	18.4–26.2 *^f^*	* ^e^ *	* ^e^ *
18	5.39 *^g^*	48.1 *^g^*	5.36–5.28 *^g^*	47.5 *^g^*
19		174.1 *^h^*		172.9 *^h^*
20	5.18 (d, *J* = 13.0 Hz, 1H) *^c^*		5.01 (d, *J* = 14.4 Hz, 1H) *^c^*	
21	2.9–3.1 *^d^*	47.2 *^d^*	* ^e^ *	* ^e^ *
22	1.2–2.3 *^f^*	18.4–26.2 *^f^*	* ^e^ *	* ^e^ *
23	1.2–2.3 *^f^*	18.4–26.2 *^f^*	* ^e^ *	* ^e^ *
24	5.45 *^g^*	47.6 *^g^*	5.67 *^g^*	46.7 *^g^*
25		173.2 *^h^*		171.6 *^h^*
26	5.23 (dd, *J* = 12.8, 3.4 Hz, 1H) *^c^*		5.36–5.28 *^c^*	
27	2.9–3.1 *^d^*	47.3 *^d^*	* ^e^ *	* ^e^ *
28	1.2–2.3 *^f^*	18.4–26.2 *^f^*	* ^e^ *	* ^e^ *
29	1.2–2.3 *^f^*	18.4–26.2 *^f^*	* ^e^ *	* ^e^ *
30	4.92–4.84 *^g^*	49.6 *^g^*	5.37 *^g^*	47.6 *^g^*
31		170.9(8)^h^		171.0(2) *^h^*
32	8.27 (d, *J* = 8.6 Hz, 1H)		8.22 (d, *J* = 9.0 Hz, 1H)	
33	5.36–5.28	49.0	5.68 (d, *J* = 5.4 Hz, 1H)	49.1
34	2.93–2.86 and 2.77–2.67	37.7	2.93–2.86 and 2.77–2.67	37.7
35		137.7(1)		137.7(4)
36	7.20–7.14 *^j^*	129.7(0)	7.20–7.14 *^j^*	129.7(3)
37	7.24 *^j^*	128.4	7.24 *^j^*	128.3
38	7.20–7.14 *^j^*	126.6(7)	7.20–7.14 *^j^*	126.6(5)
39	7.24 *^j^*	128.4	7.24 *^j^*	128.3
40	7.20–7.14 *^j^*	129.7(0)	7.20–7.14 *^j^*	129.7(3)
41		170.3		170.4

*^a^* Multiplicities, integrations and *J*-values are only shown if unambiguously characterized. *^b^*
^13^C J-modulated spin echo NMR allowed us to differentiate quaternaryC/CH_2_ from CH/CH_3_. *^c^* ^1^HNMR signals attributed to N-H from piperazate residues (N-8, N-14, N-20, N-26) could be exchangeable. *^d^* A series of overlapping ^1^HNMR signals between 2.9 and 3.1 and a series of ^13^CNMR signals between 46.9 and 47.3 are attributable to δ-methylene groups (C-9, C-15, C-21 and C-27). *^e^* Signals for methylene groups from the piperazate side chains of the minor component were not attributed due to the overlapping and/or low intensity signals. *^f^* A series of overlapping ^1^H NMR signals between 1.2 and 2.3 ppm and a series of ^13^C NMR between 18.4 and 26.2 ppm are attributable to β- and γ-methylene groups from the Piz residues (C-10, C-11, C-16, C-17, C-22, C-23, C-28, C-29). *^g^* ^1^HNMR and ^13^C NMR signals attributed to the α hydrogens and carbons from piperazate residues (C-12, C-18, C-24, C-30) could be exchangeable. *^h^* ^13^C NMR signals attributed to C=O from piperazate residues (C-13, C-19, C-25) could be exchangeable. *^i^* ^1^H NMR and ^13^C NMR signals from the imine of the dehydro-piperazate group present in compound **2**. *^j^* ^1^H NMR signals for phenylalanine aromatic carbons (C-36 to C-40) from major and minor components could not be distinguished due to overlapping.

**Table 3 ijms-24-01114-t003:** Lunaemycins produced by *S. lunaelactis* MM109^T^ and MM37.

Nb	RT (min)	Name	*m*/*z* Observed [M+H]^+^/[M+Na]^+^	Formula [M+H]^+^/[M+Na]^+^	Δm (ppm)
1	18.35	Lunaemycin A	725.4092/747.3903	C_35_H_53_N_10_O_7_^+^/C_35_H_52_N_10_O_7_Na^+^	0.1/1
2a	17.42	Lunaemycin B1	723.3935/745.3749	C_35_H_51_N_10_O_7_^+^/C_35_H_50_N_10_O_7_Na^+^	0.2/0.9
2b	17.74	Lunaemycin B2	723.3935/745.3749	C_35_H_51_N_10_O_7_^+^/C_35_H_50_N_10_O_7_Na^+^	0.2/0.9
3a	16.94	Lunaemycin C1	721.3779/743.3594	C_35_H_49_N_10_O_7_^+^/C_35_H_48_N_10_O_7_Na^+^	0.1/0.4
3b	17.27	Lunaemycin C2	721.3777/743.3594	C_35_H_49_N_10_O_7_^+^/C_35_H_48_N_10_O_7_Na^+^	0.4/0.8
4	16.82	Lunaemycin D	739.3886/761.3698	C_35_H_51_N_10_O_8_^+^/C_35_H_50_N_10_O_8_Na^+^	0.1/1
5a	15.91	Lunaemycin E1	737.3731/759.3546	C_35_H_49_N_10_O_8_^+^/C_35_H_48_N_10_O8Na^+^	0.2/0.4
5b	16.44	Lunaemycin E2	737.3731/759.3546	C_35_H_49_N_10_O_8_^+^/C_35_H_48_N_10_O_8_Na^+^	0.6/1.3
6	15.60	Lunaemycin F	741.4042/763.3856	C_35_H_53_N_10_O_8_^+^/C_35_H_52_N_10_O_8_Na^+^	0.1/0.7
7	13.76	Lunaemycin G	755.3832/777.3647	C_35_H_51_N_10_O_9_^+^/C_35_H_50_N_10_O_9_Na^+^	0.3/0.9
8	12.81	Lunaemycin H	757.399/779.3806	C_35_H_53_N_10_O_9_^+^/C_35_H_52_N_10_O_7_Na^+^	0.1/0.3
9	17.59	Lunaemycin I	711.3935/733.3751	C_34_H_51_N_10_O_7_^+^/C_34_H_48_N_10_O_7_Na^+^	0.1/0.8
10a	16.70	Lunaemycin J1	709.3784/731.3603	C_34_H_49_N_10_O_7_^+^/C_34_H_48_N_10_O_7_Na^+^	0.3/0.3
10b	17. 10	Lunaemycin J2	709.3777/731.3594	C_34_H_49_N_10_O_7_^+^/C_34_H_48_N_10_O_7_Na^+^	0.3/0.5
10c	17.40	Lunaemycin J3	709.4141/731.3951	C_35_H_53_N_10_O_7_^+^/C_35_H_52_N_10_O_6_Na^+^	0.3/0.1
Additional lunaemycin derivatives with non-elucidated structure					
11	11.73	-	753.3666/775.3492	C_35_H_49_N_10_O_9_^+^/C_35_H_48_N_10_O_9_Na^+^	1.7/0.8
12	11.79	-	771.3785/793.36	C_35_H_51_N_10_O_10_^+^/C_35_H_50_N_10_O_10_Na^+^	0.2/0.5
13	15.91	-	754.3993	C_35_H_52_N_11_O_8_^+^/ND	0.3
14	16.84	-	613.346	C_30_H_45_N_8_O_6_^+^/ND	0.4
15	18.33	-	742.436	C_35_H_56_N_10_O_7_^+^/ND	1.2
The numbers attributed to the different lunaemycins refer to the peaks displayed in EIC of Figure 7.					

**Table 4 ijms-24-01114-t004:** In vitro antibacterial activity (MIC and MBC values) of lunaemycin A on reference strains and clinical isolates with various phenotypes of resistance to antibiotics.

Organism	Resistance	MIC (µg/mL)	MBC (µg/mL)	Comparator (MIC, µg/mL) *^a^*
Gram-positive bacteria				
*Bacillus subtilis* ATCC 6633	-	0.12	4	Van, 0.5
*Enterococcus faecalis* ATCC 29212	-	0.12	8	Van, 1
*Streptococcus pyogenes* ATCC 12344	-	0.12	8	Van, 1
*Staphylococcus aureus* ATCC 25923	-	0.12	4	Van, 0.5
*Enterococcus faecalis* SI-759	Vancomycin^R^, teicoplanin^R^	0.25	8	Tei, 4
*Enterococcus faecium* SI-1831	Vancomycin^R^, teicoplanin^R^	0.25	0.25	Tei, 32
*Staphylococcus aureus* ATCC 43300	Methicillin^R^, MRSA	0.12	2	Oxa, 32
*Staphylococcus epidermidis* SI-1266	Linezolid^R^	0.12	0.25	Lin, 64
*Staphylococcus haemolyticus* SI-6/2011	Gentamicin^R^	0.12	2	Gen, >32
*Staphylococcus warneri* SI-5/2011	Oxacillin^R^	0.12	8	Oxa, 32
Gram-negative bacteria				
*Acinetobacter baumannii* ATCC 17978	*-*	>64	-	Col, 1
*Escherichia coli* CCUG^T^	*-*	>64	-	Col, 0.5
*Klebsiella pneumoniae* ATCC 13833	*-*	>64	-	Col, 0.5
*Pseudomonas aeruginosa* ATCC 27853	*-*	>64	-	Col, 1
*Acinetobacter baumannii* N50	MDR, carbapenem^R^	>64	-	Mer, 32
*Escherichia coli* SI-63	MDR, colistin^R^, mcr^+^	>64	-	Col, 8
*Escherichia coli* MO287	MDR, NDM-1^+^	>64	-	Mer, 64
*Escherichia coli* SI-1896	MDR, OXA-48-like^+^	>64	-	Mer, >16
*Klebsiella pneumoniae* B2	pandrug-resistant	>64	-	Col, 128
*Klebsiella pneumoniae* BO4	pandrug-resistant	>64	-	Col, 64
*Klebsiella pneumoniae* SI-518	MDR, NDM-1^+^	>64	-	Mer, 128
*Klebsiella pneumoniae* SI-703A	MDR, OXA-48-like^+^	>64	-	Mer, >16
*Pseudomonas aeruginosa* SI-270	MDR, colistin^R^	>64	-	Col, 16

*^a^* Van, vancomycin; Tei, teicoplanin; Oxa, oxacillin; Lin, linezolid; Gen, gentamicin; Col, colistin; Mer, meropenem. ^R^, resistant; ^+^, presence of the associated resistance gene.

**Table 5 ijms-24-01114-t005:** List of bacterial strains used in this study.

Bacteria	Ref Number	Comment/Utilization	Origin
*Streptomyces* Strains			
*S*. *lunaelactis* MM109^T^	DSM 42149	BGC 28a in pSLUN1a, type strain. Lunaemycin producer	ULiège/Hedera 22
*S*. *lunaelactis* MM37	-	BGC 28a in pSLUN1a. Lunaemycin producer	ULiège/Hedera 22
*S*. *lunaelactis* MM22	-	BGC 28b in pSLUN1b	ULiège/Hedera 22
*S*. *lunaelactis* MM25	-	BGC 28b in pSLUN1b	ULiège/Hedera 22
*S*. *lunaelactis* MM28	-	BGC 28b in pSLUN1b	ULiège/Hedera 22
*S*. *lunaelactis* MM31	-	BGC 28b in pSLUN1b.	ULiège/Hedera 22
*S*. *lunaelactis* MM40	-	BGC 28b in pSLUN1b	ULiège/Hedera 22
*S*. *lunaelactis* MM78	-	BGC 28b in pSLUN1b	ULiège/Hedera 22
*S*. *lunaelactis* MM83	-	BGC 28b in pSLUN1b	ULiège/Hedera 22
*S*. *lunaelactis* MM113	-	BGC 28b in pSLUN1b	ULiège/Hedera 22
*S*. *lunaelactis* MM115	-	BGC 28b in pSLUN1b	ULiège/Hedera 22
Strains used for antibacterial assays			
*Acinetobacter baumannii*	ATCC 17978	Reference strain	ULiège
*Acinetobacter baumannii*	N50	Clinical isolate (MDR, carbapnem^R^)	Università di Siena
*Bacillus subtilis*	ATCC 6633	Reference strain	Università di Siena
*Bacillus subtilis*	ATCC 19659	Reference strain	ULiège
*Citrobacter freundii*	ATCC 43864	Reference strain	ULiège
*Escherichia coli*	CCUG^T^	Reference strain	Università di Siena
*Escherichia coli*	ATCC 25922	Reference strain	ULiège
*Escherichia coli*	SI-63	Clinical isolate (MDR, colistin^R^, mcr^+^)	Università di Siena
*Escherichia coli*	MO287	Clinical isolate (MDR, NDM-1^+^)	Università di Siena
*Escherichia coli*	SI-1896	Clinical isolate (MDR, OXA-48-like^+^)	Università di Siena
*Enterococcus faecalis*	ATCC 29212	Reference strain	Università di Siena
*Enterococcus faecalis*	SI-759	Clinical isolate (vancomycin^R^, teicoplanin^R^)	Università di Siena
*Enterococcus faecium*	SI-1831	Clinical isolate (vancomycin^R^, teicoplanin^R^)	Università di Siena
*Enterococcus hirae*	ATCC 9790	Reference strain	ULiège
*Klebsiella pneumoniae*	ATCC 13833	Reference strain	ULiège/Università di Siena
*Klebsiella pneumoniae*	B2	Clinical isolate (pandrug-resistant)	Università di Siena
*Klebsiella pneumoniae*	BO4	Clinical isolate (pandrug-resistant)	Università di Siena
*Klebsiella pneumoniae*	SI-518	Clinical isolate (MDR, NDM-1^+^)	Università di Siena
*Klebsiella pneumoniae*	SI-703A	Clinical isolate (MDR, OXA-48-like^+^)	Università di Siena
*Kocuria rhizophila* (*Micrococcus luteus*)	ATCC 9341	Reference strain	ULiège
*Proteus mirabilis*	ATCC 7002	Reference strain	ULiège
*Pseudomonas aeruginosa*	ATCC 27853	Reference strain	ULiège/Università di Siena
*Pseudomonas aeruginosa*	SI-270	Clinical isolate (MDR, colistin^R^)	Università di Siena
*Serratia marcescens*	ATCC 10759	Reference strain	ULiège
*Streptococcus pyogenes*	ATCC 12344	Reference strain	ULiège
*Staphylococcus aureus*	ATCC 25923	Reference strain	ULiège/Università di Siena
*Staphylococcus aureus*	ATCC 43300	Clinical isolate (methicillin^R^, MRSA)	Università di Siena
*Staphylococcus epidermidis*	SI-1266	Clinical isolate (linezolid^R^)	Università di Siena
*Staphylococcus haemolyticus*	SI-6/2011	Clinical isolate (gentamicin^R)^	Università di Siena
*Staphylococcus warneri*	SI-5/2011	Clinical isolate (oxacillin^R)^	Università di Siena

^R^, resistant; ^+^, presence of the associated resistance gene.

## Data Availability

The Molecular Networking job in GNPS can be found at https://gnps.ucsd.edu/ProteoSAFe/status.jsp?task=84ea45db68814c94a3b2bad9c26dd807, accessed on 14 October 2019.

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
