# Peer review of "Lunaemycins, New Cyclic Hexapeptide Antibiotics from the Cave Moonmilk-Dweller *Streptomyces lunaelactis* MM109^T^"

_ijms, 2023, doi:10.3390/ijms24021114_

Round 1
Reviewer 1 Report
The manuscripts brings an interesting proof how, in certain type of metabolites produced by modular systems and not much further modified, the thorough bioinformatic inspection of genomic data may assist with the compounds identification. The work is well-designed and the findings are persuasive.
There are only few minor comments to misprints and formatting:
1. In the Table 1 please check the uniformity in capitalization of the statements of the gene homologies and proposed functions collumns
2. In the 491 line, there is a double “ee” in lunaemycin
As for the results presented:
1. In the Table 4 – in majority of the Gram-positives, the effect of the lunaemycin is rather bacteriostatic (MBC >>MIC]. However, in some resistant strains of enterococci and staphylococci their effect changes to bactericidal (MIC is identical or almost identical with MBC). Is it a common finding for well-established antibiotics to behave this way in resistant clinical isolates? Is there any possible explanation of such phenomenon? The Enterococcus faecium SI-1831 is vanR teiS, so perhaps it is not VanA mutant, but rather VanB or any other. On the other hand, in Staphylococcus epidermidis SI-1266 linezolid resistance most probably relates to a point mutations of ribosomal rRNA or L3 and L4 proteins as reported in other linezolid-resistant staphylococci. Why different resistance mechanisms increase bactericidal potency of lunaemycin? Could you add a short comment on this, please?
2. In the 3.3., please add more details on how the extracts were prepared.
3. The results indicate that lunaemycins show attractive anti-Gram+ activities. However, some structurally related compounds are anti-proliferative, too. Have you assayed cytotoxicity and anti-proliferative features of the compounds?
Reviewer 2 Report
Martinet et al describe the discovery of new cyclic hexapeptides with antimicrobial activity from a cave-dwelling streptomycete. Their approach stands out through the unusual and secluded habitat of this organism and their use of different strains of the same organism to facilitate product identification.
The paper is very well-written and the story is told in a clear and easily understandable way. The figures are very well-designed, aesthetically appealing, helpful, and clearly intelligible. The supplementary material is appropriate and of the same quality as the main display items.
I only have a few comments to be addressed prior to publication (in order of appearance in the manuscript):
Figure 2: Some items are very small and hard to read.
Figures 2, 4, 6: These figures appear of low resolution. This may be due to the template for review, but please make sure that in the final version, high-resolution figures are used.
Table 2: The thick grey bar in the middle is quite distracting and unnecessary. A simple line would do better.
Fig 5C: I would wish a bit more explanation as to how to interpret this figure. Having no experience with COSY and HMBC, I could not follow the authors’ reasoning in the text.
Figure 8: What was the reasoning to use of gentamycin as positive control? It does not appear to be active against several of these strains. Should the point of a positive control not be that it shows that the assay worked? Please consider using different positive controls for the strains where gentamycin shows no activity.
Table 4: It could be beneficial to include one or two established antibiotics here for comparison, just to see how the lunaemycins perform compared to currently used antibiotics. Maybe daptomycin or vancomycin for the Gram-positive strains? Since there is no activity against Gram-negative strains, this control would be obsolete there.
